# Role of Density and Grain Size on the Electrocaloric Effect in Ba_0.90_Ca_0.10_TiO_3_ Ceramics

**DOI:** 10.3390/ma15217825

**Published:** 2022-11-06

**Authors:** Lavinia Curecheriu, Maria Teresa Buscaglia, Vlad Alexandru Lukacs, Leontin Padurariu, Cristina Elena Ciomaga

**Affiliations:** 1Dielectrics, Ferroelectrics and Multiferroics Group, Faculty of Physics, Al. I. Cuza University of Iasi, 11 Carol I Bv., 700506 Iasi, Romania; 2Institute of Condensed Matter Chemistry and Technologies for Energy, National Research Council-CNR, Via De Marini 6, Genoa I-16149, Italy; 3Department of Exact and Natural Sciences, Institute of Interdisciplinary Research, Al. I. Cuza University, 11 Carol I Bv., 700506 Iasi, Romania

**Keywords:** Ca-BaTiO_3_, porosity, grain size, electrocaloric effect

## Abstract

Pure perovskite Ba_0.90_Ca_0.10_TiO_3_ ceramics, with a relative density of between 79 and 98% and grain sizes larger than 1 µm, were prepared by solid-state reaction. The dielectric and electrocaloric properties were investigated and discussed considering the density and grain size of the samples. Room temperature impedance measurements show good dielectric properties for all ceramics with relative permittivity between 800 and 1100 and losses of <5%. Polarization vs. E loops indicates regular variation with increasing sintering temperature (grain size and density), an increase in loop area, and remanent and saturation polarization (from P_sat_ = 7.2 µC/cm^2^ to P_sat_ = 16 µC/cm^2^). The largest electrocaloric effect was 1.67 K for ceramic with GS = 3 µm at 363 K and electrocaloric responsivity (ζ) was 0.56 K mm/kV. These values are larger than in the case of other similar materials; thus, Ba_0.90_Ca_0.10_TiO_3_ ceramics with a density larger than 90% and grain sizes of a few µms are suitable materials for electrocaloric devices.

## 1. Introduction

The electrocaloric effect (EC) is an adiabatic temperature change (ΔT) to an externally applied field due to the coupling of electrical and thermal properties [1]. In recent years, it has become a challenging research topic in the field of ferroelectric materials due to their possible application as solid-state cooling devices. The largest EC effects were discovered in some lead-based ferroelectric films (ΔT = 12 K in a Pb(Zr,Ti)O_3_ film [2]); however, because of international limitations on lead use, another challenge was to develop lead-free EC materials. BaTiO_3_ (BT)-based compounds are attractive because they are environmentally friendly and have large thermal stability. Moreover, in BT-based systems, various parameters (transition temperature, dielectric permittivity, piezoelectric and pyroelectric coefficient, and ferroelectricity) can be tuned by suitable doping [3]. It is known that doping either at a Ba or Ti site affects transition temperatures and offers the possibility to close it to room temperature [4]. This aspect is being utilized by many research groups to enhance the EC effect in ferroelectric materials because EC has a maximum near-phase transition as compared to other temperature regions [5]. For example, Jian et al. have reported ΔT = 2.4 K in BaZr_0.05_Ti_0.95_O_3_ at 113 °C and E = 30 kV/cm [6]; Bai et al. reported ΔT = 1.6 K in BT single crystal at T_C_ and E = 10kV/cm [7]; Niu et al. reported ΔT = 2.42 K and ΔT = 2.46 for Sr-doped BaTiO_3_ (0.35 and 0.40) at E = 50 kV/cm [8]; in a double substitute BaTiO_3_ (Ba_0.85_Ca_0.15_Ti_0.94_Hf_0.06_O_3_), Wang et al. reported ΔT = 1.03 K at E = 35 kV/cm [9].

Doping BT with Ca^2+^ (Ba_1−x_Ca_x_TiO_3_-BCT) is one of the foremost potential candidates for lead-free electrooptic modulators [10]. Few papers have reported the microstructure and dielectric properties of BCT ceramics prepared by solid-state reactions [11,12,13,14]. Early papers [11,12] revealed that Ca^2+^ replaces Ba^2+^ with a solubility up to 0.21 causing a small change in Curie point but strongly lowering the tetragonal-orthorhombic transition temperature, which increases thermal stability, and it is important for many practical applications. An interesting phenomenon is that a small amount of Ca^2+^ ions can substitute Ti^4+^ if the solid-state technique is used [15]. In this case, the change in temperature transition is completely different, and a crossover to a relaxor state can appear. If the substitution is only in the Ba^2+^ site, the Ca ions might have greater atomic polarizability, which intensifies the interactions with Ti ions [16].

Although some papers were published concerning dielectric and ferroelectric properties of BCT ceramics [10,11,12,13,14,15,16], only recently have the electrocaloric effect properties of these ceramics been investigated, and mainly for the double substituted BaTiO_3_ [17,18,19]. The only paper that reported electrocaloric properties in Ca-doped BT is that of AS Anokhin et al. [18]. The authors investigated the EC effect in 10% Ca-doped BaTiO_3_ and obtained a maximum of EC at the transition temperature. Although that paper has no data about the microstructure properties of the samples and presented only this composition, the authors conclude that BCT ceramics are more suitable for electrocaloric devices than BT.

Starting from these results, in the present paper, we proposed a systematic study of the role of porosity and grain size on the electrocaloric properties of Ba_0.90_Ca_0.10_TiO_3_ (BCT) ceramics. The paper reports, for the first time, BCT ceramics with large grain size distribution (from one µm to tens of µms) and porosity between 21 and 2%, and discussed the influence of these microstructural particularities on dielectric and electrocaloric properties.

## 2. Materials and Methods

Ba_0.90_Ca_0.10_TiO_3_ (BCT) nanopowders have been prepared by a classical ceramic method using the following chemical reaction:0.90 BaCO_3_ + 0.10 CaCO_3_ + TiO_2_ → Ba_0.90_Ca_0.10_TiO_3_ + CO_2_↑

BaCO_3_ (Solvay, 99.9% purity, București, Romania), TiO_2_ (Degussa-P25, 99.9% purity, București, Romania), and CaCO_3_ (Solvay 99.9%, București, Romania) nanopowders were weighted in stoichiometric proportions, then wet-mixed in distilled water for 24 h. After freeze-drying, the powders were calcinated at 900 °C for 4 h. The final powders were isostatically cold pressed at 1400 bar. The ceramic greens were sintered for 4 h in the air at different temperatures between 1300 °C and 1450 °C to induce different porosity levels and grain sizes. The density of ceramics was obtained by Archimedes’ method. The ceramics were noted according to their porosity as shown in Table 1.

Phase symmetry of the sintered ceramics was verified using X-ray diffraction (XRD) with CuKα radiation (Shimadzu LabX 6000 diffractometer, Shimadzu, București, Romania). The microstructure was observed through high-resolution scanning electronic microscopy LEO 1450VP (Carl Zeiss, București, Romania). For electrical measurements, Ag electrodes were deposited on the polished surfaces of the ceramics. The room temperature dielectric measurements were carried out using the Solartron 1260 (Solartron Analytical, Hampshire, UK) for frequencies ranging from 1 Hz to 1 MHz. The electrocaloric effect was indirectly determined from P(E) loops registered at different temperatures. The measurements were performed on ceramic disks immersed in a silicon oil bath using the Radiant Precision Multiferroic II Ferroelectric Test System (Radiant Technologies, INC., Albuquerque, NM, USA) at 10 Hz with a double bipolar input as the electric signal.

## 3. Results and Discussion

### 3.1. Structural and Microstructural Characteristics

Figure 1 shows the XRD patterns of the BCT ceramics at a few selected porosity levels (21%, 7%, and 2%). The lack of any Ba, Ca or Ti-rich secondary phases in the XRD detection limit demonstrates that the solid-state reactions took place, and the Ca ions are completely incorporated in the perovskite lattice. All diffraction peaks can be indexed based on polycrystalline orthorhombic BaTiO_3_ (01–081–2197) with the space group Amm2. Unlike our previous papers [20,21], in which porosity induced structural changes, in this case, all the ceramics are in the orthorhombic phase at room temperature with a small variation in peak ratio (022)/(200) from 1.39 to 1.14 when porosity decreased from 21% to 2%.

Figure 2 presents the SEM micrographs performed on freshly fractured ceramics revealing the microstructural features of BCT ceramics. While ceramics sintered at low temperatures (<1375 °C) present irregular pores and grain sizes between 1 and 3 µm, the ceramics obtained at high temperatures (>1400 °C) are fully densified with a small intergranular porosity and grain sizes of tens of µm. For the ceramics with small grains, the insets in the bottom right corner of the SEM images (Figure 2a–c) show the histograms corresponding to the GS distribution and the average GS. In the case of ceramics with larger grains, the size distribution could not be carried out; however, we can conclude that by increasing the sintering temperature, the porosity becomes very small, pores are well isolated, and grains increase to tens of µm. This kind of porosity is similar to that one obtained for Ba(Zr, Ti)O_3_ ceramics [22] and is due to the sintering strategy. By classical sintering method (cold isostatic press and thermal treatment) it is difficult to achieve good densification and small grain size [23]. However, unlike BZT ceramics, these ceramics have a homogeneous microstructure without bimodal grain size distribution.

### 3.2. Low Field Properties

The frequency dependence of the permittivity and dielectric losses at room temperature are shown in Figure 3a,b. They indicate an increase in relative permittivity with density and grain size from 815 for BCT21 to 1100 for BCT2 at a fixed frequency of f = 10 kHz. This result can be regarded as a sum property, a direct proportion between the permittivity value and the amount of the ferroelectric phase. The samples present a small dispersion in frequency (~12% difference between permittivity at 1 Hz and permittivity at 1 MHz) except ceramic BCT8. This sample has a difference in permittivity values of ~22% at 1 Hz and 1 MHz. Furthermore, this ceramic presents the largest dielectric loss for *f* of <103 Hz. An interesting feature of this ceramic is the comparable value of permittivity with BCT7, which has an exaggerated grain growth. The dielectric losses of all ceramics are smaller than 5% of all investigated frequencies without any relaxation.

### 3.3. Ferroelectric Properties

The P(E) polarization-field loops in the dynamic ac regime have been first recorded at room temperature to assess the role of porosity and grain size on the switching properties of BCT ceramics. The P(E) major loops represented in Figure 4 indicate that porosity causes a strong reduction in remanent and saturation polarization, together with a loop tilting that accompanies the reduction in hysteresis area and P_rem_/P_sat_ rectangularity loop factor (Table 1). The maximum polarization is in the range of 7.2–16 μC/cm^2^, with a remanent polarization in the range of 2–9.9 μC/cm^2^, as shown in Table 1. Both results are in good agreement with the other literature data reported for this system [24]. The ferroelectric behavior seems to be affected only by porosity. In our previous papers, we demonstrated by Finite Element Calculation [25,26] that the electric field acting on the active material in porous ceramics is inhomogeneous and smaller than the applied one, and this determined the reduction in rectangularity and tilting P(E) loops.

### 3.4. Electrocaloric Properties

The EC effect was investigated in lead-free BCT ceramics for assessing the suitability of eco-friendly solid-state cooling devices. Two approaches to obtaining the adiabatic temperature change (ΔT) can be used. One is called the indirect method, which used Maxwell relations and the polarizations extracted from P(E) loops at different temperatures and electric fields, and the other is called the direct method, which measures the temperature change when applied to an external electric field in an adiabatic condition. In this paper, we used the indirect method.

Figure 5 shows the temperature dependence of the ferroelectric hysteresis loop for a few selected samples, which are BCT11, BCT8, BCT3, and BCT2. For all the samples, the loops are well saturated, and the remanent polarization decreases with the increase in temperature, and finally, the loops turn into a slimmer loop (paraelectric state) for all porosities. In the insets of Figure 5, we present P(T) under different electric fields from 30 kV/cm up to 0 kV/cm. It can be observed that the polarization decreases as a function of temperature followed by an abrupt drop at Curie temperature for all the ceramics irrespective of grain size or porosity. A particular behavior is in the case of BCT8, which presents a small increase in polarization for temperatures between 300 and 320 K.

To calculate the adiabatic electrocaloric temperature change (ΔT), the Maxwell relation was used:(1)$$\Delta T\;=-\frac{T}{{\rho C}_{E}}\int_{E_{1}}^{E_{2}}{\left(\frac{\partial P}{\partial T}\right)}_{E}\mathrm{dE}$$
where ρ is the density of the ceramics, C_E_ is the specific heat capacity at a fixed field, and E_1_ and E_2_ represent the limits for the applied electric fields. The values for the specific heat of the dense sample are taken from the literature (C_E_ = 0.40 J/gK) [27], and the porous ceramics are calculated from [28]:(2)$${C^{\prime }}_{E}={\;C}_{E}x\beta$$
(3)$$\beta \;=1-\frac{\rho_{\mathrm{porous}}}{\rho_{\mathrm{dense}}}$$
because the specific heat of pores is less than dense materials, resulting in a decrease in the specific heat of porous ceramics. Figure 6 shows ΔT as a function of temperature for all BCT ceramics at a few selected applied fields. Irrespective of porosity or grain size, all the samples have the maximum ΔT around the ferroelectric–paraelectric phase transition temperature. Additionally, a few remarks can be made: (i) by increasing the applied field, the maximum ΔT shifts to higher temperatures because the transition temperatures also shift to higher temperatures with fields, as already shown in our previous papers [21,29]; (ii) BCT8 and BCT2 present small peaks around between 300 and 320 K that correspond to peaks in P(T) dependences. These are not field-dependent and cannot be attributed to a structural phase transition; (iii) BCT8, BCT7, and BCT3 have the largest values of ΔT (almost the same). Considering that these ceramics have a variation of density of 5% and grain size from 3 µm to 10 µm, it seems that the effect of porosity was canceled by the grain sizes.

To compare our results with similar systems reported in the literature, the electrocaloric responsivity (ζ = ΔT/ΔE) was calculated and the results are presented in Figure 7 for all samples. In our case, the maximum of ζ is 0.56 K mm/kV, which is much higher than the 0.49 K mm/kV reported for the same composition by the direct method [18] and 0.17 K mm/kV for Ba_0.8_Ca_0.2_TiO_3_ [19]. This result is also very good in comparison with another barium titanate solid solution, single or double substituted [30,31,32].

## 4. Conclusion

Ba_0.90_Ca_0.10_TiO_3_ ceramics with a porosity between 21% and 2% and grain sizes from 1.5 µm–10s of µm were prepared by solid-state reaction. The samples present very good room temperature dielectric properties with permittivity larger than 800 and dielectric losses below 5%. All the samples have well-saturated hysteresis loops with a regular increase in loop area and remanent and saturation polarization with density and grain sizes. The maximum electrocaloric responsivity was 0.56 Kmm/kV obtained for three different samples with densities of approximately 93% r.d and grain sizes larger than three µm. This response is larger than others reported in the literature for similar materials. In summary, we can conclude that Ba_0.90_Ca_0.10_TiO_3_ ceramics with a density larger than 90% and grain sizes of a few µm are suitable for electrocaloric devices.

## Figures and Tables

**Figure 1 materials-15-07825-f001:**
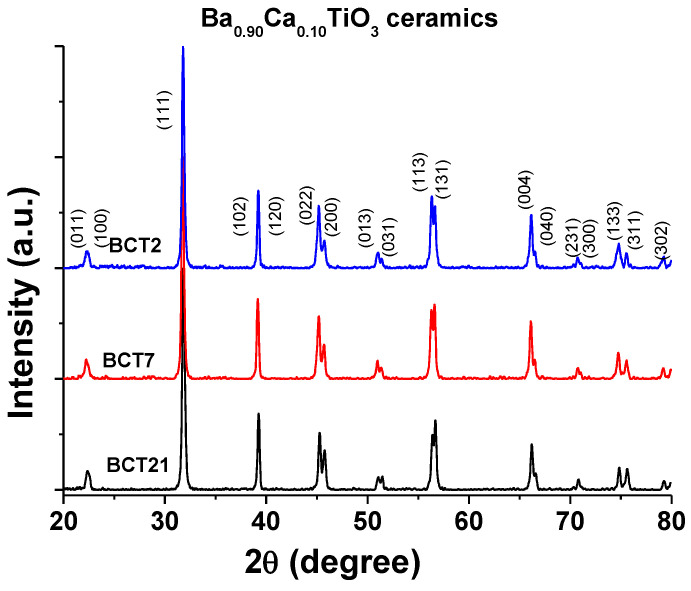
Room temperature XRD of a few selected BCT ceramics.

**Figure 2 materials-15-07825-f002:**
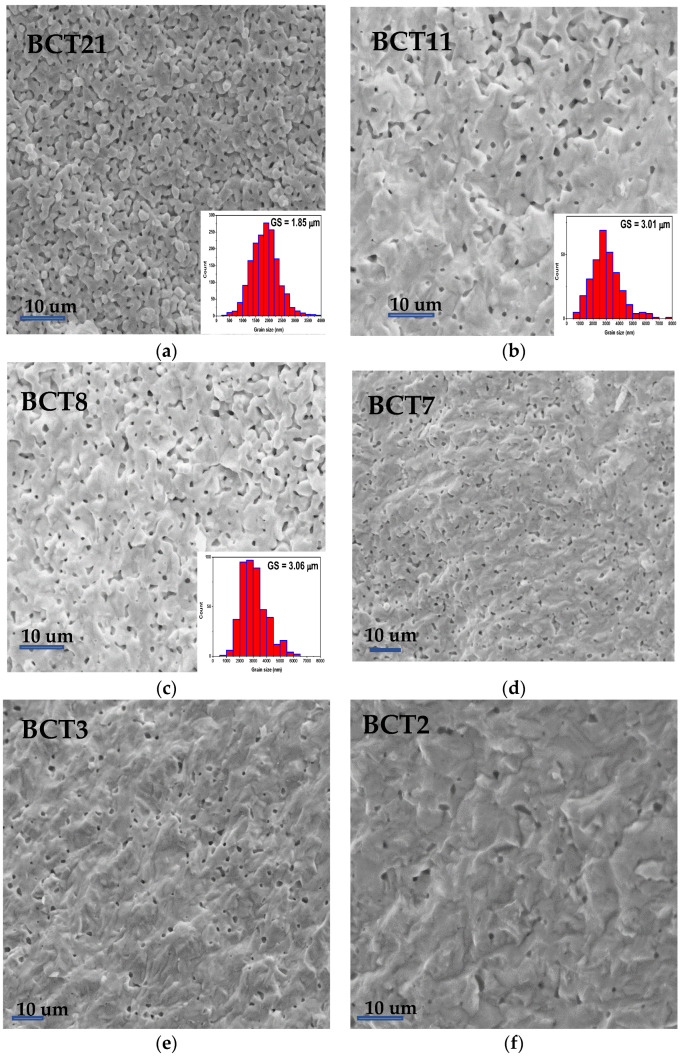
SEM images of Ba_0.90_Ca_0.10_TiO_3_ ceramics with different porosity levels and grain size: (**a**) 21% porosity; (**b**) 11% porosity; (**c**) 8% porosity; (**d**) 7% porosity; (**e**) 3% porosity; (**f**) 2% porosity.

**Figure 3 materials-15-07825-f003:**
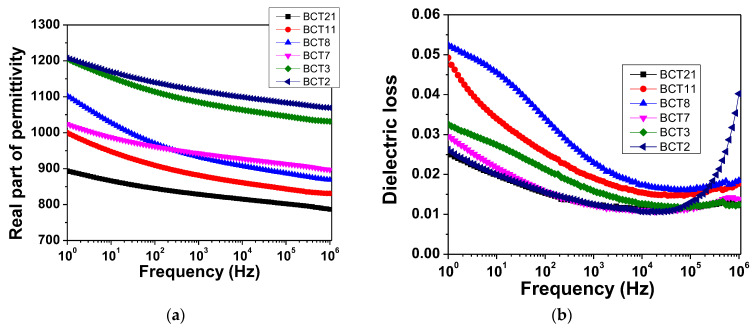
Frequency dependence of the real part of permittivity (**a**) and dielectric loss (**b**) at room temperature for Ba_0.90_Ca_0.10_TiO_3_ ceramics with different porosity levels.

**Figure 4 materials-15-07825-f004:**
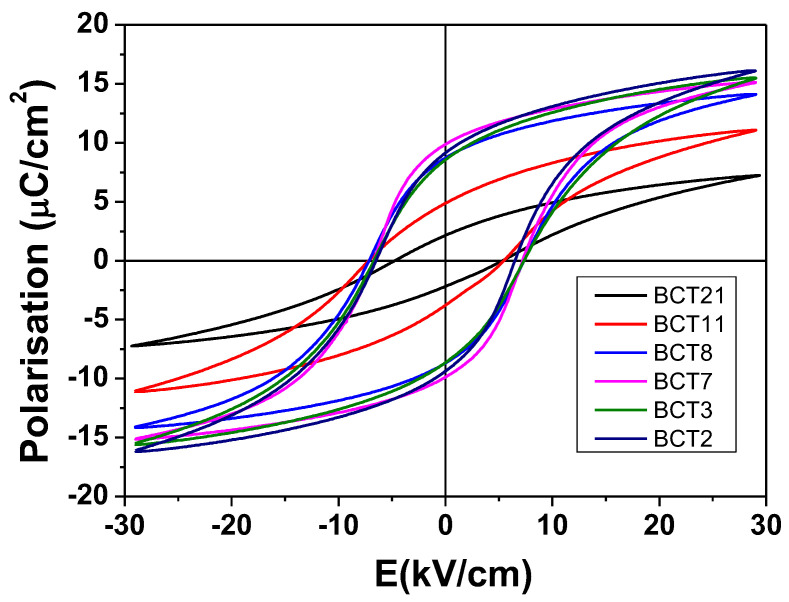
Room temperature P€ hysteresis loops for Ba_0.90_Ca_0.10_TiO_3_ ceramics.

**Figure 5 materials-15-07825-f005:**
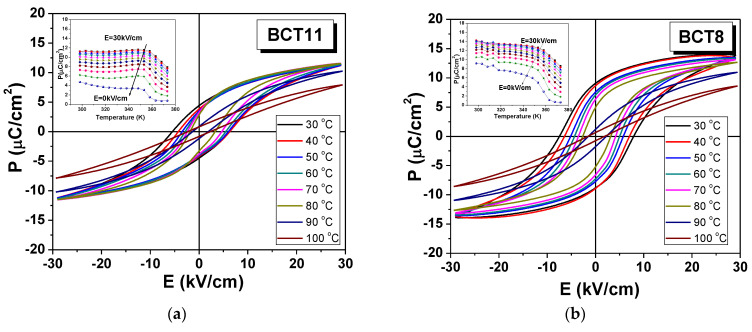
P(E) loops at a few selected temperatures for Ba_0.90_Ca_0.10_TiO_3_ ceramics with different porosity levels: (**a**) 11%, (**b**) 8%, (**c**) 3%, and (**d**) 2% porosity.

**Figure 6 materials-15-07825-f006:**
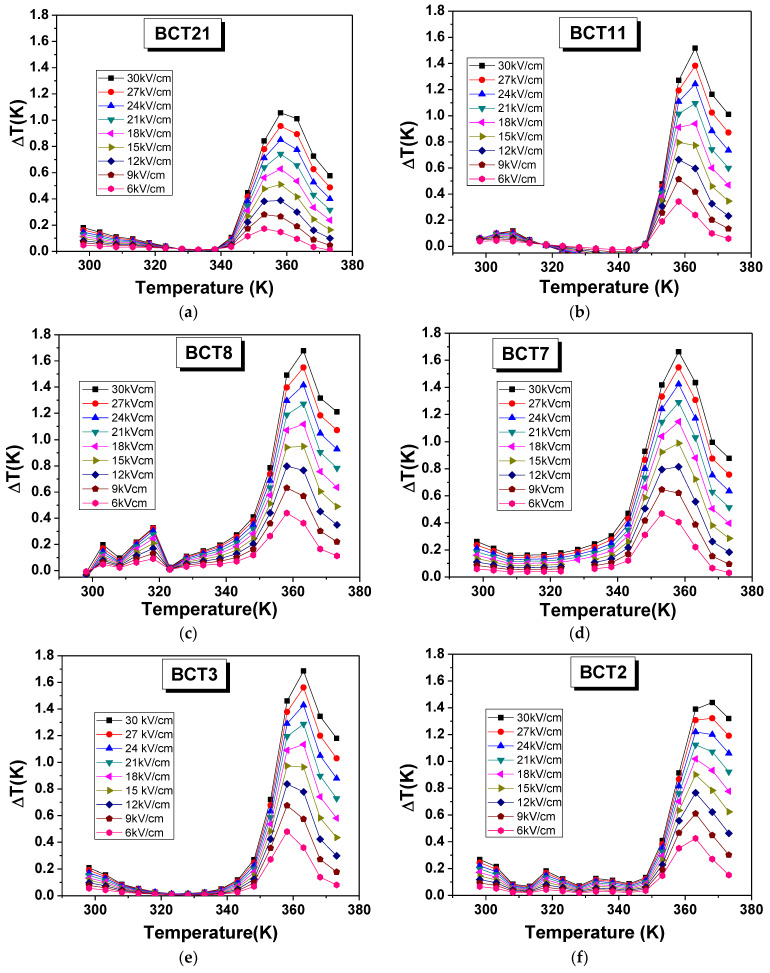
Electrocaloric adiabatic temperature change as a function of temperature at a few selected applied fields: (**a**) BCT21, (**b**) BCT11, (**c**) BCT8, (**d**) BCT7, (**e**) BCT3, and (**f**) BCT2.

**Figure 7 materials-15-07825-f007:**
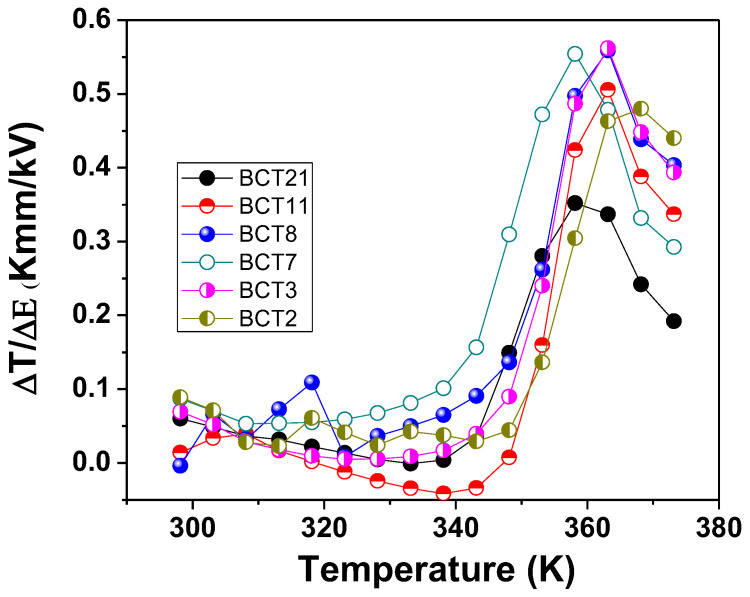
Electrocaloric responsivity as a function of temperature.

**Table 1 materials-15-07825-t001:** Sample notation and dielectric and ferroelectric room temperature characteristics of Ba_0.90_Ca_0.10_TiO_3_ ceramics.

Sample	Thermal Treatment (°C/4 h)	Relative Density (%)	ε (f = 10 kHz)	P_sat_ (µC/cm^2^)	P_rem_ (µC/cm^2^)	P_rem_/P_sat_
BCT21	1300	79	815	7.21	2.17	0.30
BCT11	1350	89	861	11.08	4.88	0.44
BCT8	1375	92	907	14.10	8.73	0.62
BCT7	1400	93	927	15.09	9.89	0.65
BCT3	1425	97	1063	15.5	8.53	0.55
BCT2	1450	98	1099	16.11	9.19	0.57

## Data Availability

Data will be made available on request.

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
