# Peer review of "Role of Density and Grain Size on the Electrocaloric Effect in Ba0.90Ca0.10TiO3 Ceramics"

_materials, 2022, doi:10.3390/ma15217825_

Round 1

Reviewer 1 Report

The authors have addressed the Role of density and grain size on the electrocaloric effect in Ba0.90Ca0.10TiO3 ceramics. After carefully reviewing the manuscript, I noticed that the work had been well carried out, and the results are fascinating. The authors have done a systematic analysis, and the investigation is helpful for the scientific community. I would recommend it be published after addressing the following two comments:

1.     Better define the novelty and importance of the work. Be clear in stating what aspects of this work are new, what results were interesting or surprising, and how this work will impact future research and development in this field.

2.      In the overall manuscript, spell check, spacing, Figure adjustment, and typing errors are to be checked.

Author Response

Dear reviewer,

Thank you very much for reading and correcting our paper.

  1. We pointed out both in the introduction part and the abstract the novelty of the paper. 
  2. The manuscript was checked again, and we hope the English was improved. 

Reviewer 2 Report

The comments are presented in the attached file.

Author Response

Dear reviewer,

Thank you for the revision of our manuscript. 

Please find in the following the list of all changes that were made to your comments.

  1. We added the corresponding authors in the manuscript.
  2. We rephased the sentence. The values are for relative permittivity at room temperature and a fixed frequency of 10 kHz. 
  3. We corrected the caption of the figure. All the samples are the same composition: Ba0.90Ca0.10TiO3. The percentage represents the porosity level. 
  4. All the ceramics were sintered at the same rate: 50C/min both at the increase and the decrease temperature in order to eliminate the effect of the sintered rate. So, in this case, we can conclude that the differences in the electrocaloric effect are due to only porosity and grain sizes. 

Reviewer 3 Report

In this report, authors prepared perovskite-structured Ba0.9Ca0.1TiO3 with different densities and grain sizes by solid-state reaction. Their electrocaloric properties were evaluated and the effects of samples' density & grain size to their properties were studied. Generally I recommend to accept this report after minor revision for publication on Materials. Some specific comments are listed below:

1. Authors suggested the success of doping Ca in BaTiO3 lattice, as no other impurity phases were observed in pXRD patterns. Note that lab-base pXRD usually has some limits on seeing very small amount of components or some details of the detected samples. So if available, Rietveld refinement on the obtained pXRD patterns (at least for a representative one) is recommended. Another solution can be to use EDX in SEM, which should be able to confirm the uniform distribution of elements in elementary mapping, and give a semi-quantitative composition.

2. Reference [27] is somehow highlighted and looks incomplete in the manuscript, which may need to be fixed.

Author Response

Dear reviewer,

Thank you for the revision of our manuscript. 

Please find in the following the list of all changes that were made to your comments.

  1. We pointed out in the manuscript that the phase purity is in the XRD limit. Unfortunately, we cannot perform Rietveld refinement but, we used the EDX from SEM for ceramic BCT7. We checked the composition in different points and the Ba, Ca, Ti are almost uniformly distributed in ceramics. The small amounts of Si and Zn, observed in some points are more probably related to the polishing process of the ceramic.
  2. We fixed references [27].

The results are presented in word file. 

Reviewer 4 Report

The authors report electrocaloric responsivity of 0.56 Kmm/kV for Ba0.90Ca0.10TiO3 ceramics.  The results are promising and well stated.

Minor English spelling are required.

Author Response

Dear reviewer,

Thank you very much for reading and correcting our paper. The manuscript was checked again, and we hope the English was improved.